# Transfer Learning for Image-Based Malware Detection for IoT

**DOI:** 10.3390/s23063253

**Published:** 2023-03-20

**Authors:** Pratyush Panda, Om Kumar C U, Suguna Marappan, Suresh Ma, Manimurugan S, Deeksha Veesani Nandi

**Affiliations:** 1School of Computer Science and Engineering, Vellore Institute of Technology, Chennai 600127, India; 2Amrita School of Business, Amrita Vishwa Vidyapeetham, Coimbatore 641112, India; 3Faculty of Computers and Information Technology, University of Tabuk, Tabuk 71491, Saudi Arabia; 4Technical Lead, Virtusa Consulting Services, Chennai 603103, India

**Keywords:** malware detection, CNN, transfer learning, ensemble, autoencoder, GRU, MLP, MalImg

## Abstract

The tremendous growth in online activity and the Internet of Things (IoT) led to an increase in cyberattacks. Malware infiltrated at least one device in almost every household. Various malware detection methods that use shallow or deep IoT techniques were discovered in recent years. Deep learning models with a visualization method are the most commonly and popularly used strategy in most works. This method has the benefit of automatically extracting features, requiring less technical expertise, and using fewer resources during data processing. Training deep learning models that generalize effectively without overfitting is not feasible or appropriate with large datasets and complex architectures. In this paper, a novel ensemble model, Stacked Ensemble—autoencoder, GRU, and MLP or SE-AGM, composed of three light-weight neural network models—autoencoder, GRU, and MLP—that is trained on the 25 essential and encoded extracted features of the benchmark MalImg dataset for classification was proposed. The GRU model was tested for its suitability in malware detection due to its lesser usage in this domain. The proposed model used a concise set of malware features for training and classifying the malware classes, which reduced the time and resource consumption in comparison to other existing models. The novelty lies in the stacked ensemble method where the output of one intermediate model works as input for the next model, thereby refining the features as compared to the general notion of an ensemble approach. Inspiration was drawn from earlier image-based malware detection works and transfer learning ideas. To extract features from the MalImg dataset, a CNN-based transfer learning model that was trained from scratch on domain data was used. Data augmentation was an important step in the image processing stage to investigate its effect on classifying grayscale malware images in the MalImg dataset. SE-AGM outperformed existing approaches on the benchmark MalImg dataset with an average accuracy of 99.43%, demonstrating that our method was on par with or even surpassed them.

## 1. Introduction

The relevance of cybersecurity has grown significantly as a result of the increased reliance on computer systems, the Internet, wireless networks, and the expansion of smart devices that make up the IoT. The term “cybersecurity” [1] describes the process of defending our computer networks and systems against unauthorized access, theft, and damage to electronic data, hardware, or software, as well as against service interruption or rerouting. Cybersecurity is a serious worry due to the complexity of information systems today. The major goal is to ensure that the system is trustworthy and that all its components are intact. Numerous cyber-attacks, such as denial of service (DoS), distributed denial of service (DDoS), direct-access attacks, phishing, eavesdropping, reverse engineering, side-channel attacks, malware attacks, etc., occur every second.

With the shift of focus to cloud computing, there is a major concern regarding the privacy of its users [2,3]. A distributed cloud storage approach using an encryption technique to prevent the data mining attacks launched on a single cloud storage system was proposed in [4], which increased the workload of an attacker, thereby increasing privacy and ensuring reliability. A spike in IoT botnet flash attacks on social networking sites and IT industries were observed, which were addressed with the use of machine and deep learning techniques ([5,6,7,8]). Even though IoT paved our way in development, it also opened up an opportunity for attackers to use botnets from remote locations to invade our privacy and snatch our data. Any kind of attack mentioned previously is possible using these botnets. Different types of IDS, or intrusion detection systems, were developed to investigate any suspicious activity or network traffic to alert us. Many authors implemented IDS using machine and deep learning algorithms with the objective of not only identifying suspicious activities but also taking the appropriate measures to reduce or eliminate those attacks [9,10,11,12,13].

Malware assaults have recently dominated all other attacks in terms of frequency and severity. Malicious programs or pieces of code that disrupt computer networks, allow illegal access, deny access to information, reveal private information, etc., as well as interfere with security and privacy, fall under the general term ‘malware’ [14], often known as ‘malicious software’. Trojan-horse, worms, viruses, spyware, ransomware, etc. are some examples of well-known malware. Every home practically has at least one gadget that might be infected by malware, and the number of malware and their varieties is growing daily. By the end of 2020, 5.22 billion people worldwide—or 66% of the world’s population—were smartphone users (DataReportal, 2021) [15]. A 1.8% rise over the year-end total for 2019 was achieved in 2020 with the addition of 93 million users. The most common OS for mobile devices was found to be Android [16]; in fact, it was used in many devices, including smartphones, tablets, wearables such as smartwatches and augmented reality headsets, and mobile devices such as smartphones and wearables. Given the widespread use of the Android OS and the fact that it is open-source, along with the sensitive data we constantly keep on our mobile devices, rogue code authors create ever more aggressive codes every day intending to steal our data [17,18,19].

Malware has become a dynamic ecology because of the ongoing tension between security experts and hackers. Despite the long-term tendencies that can be seen in the reports comparing year over year, changes in the malware ecosystem occur every year. Although there are many anti-virus mechanisms in place, hackers and cybercriminals never give up easily, particularly if there is profit from infection. As hackers adapt their strategies to target fresh or seldom-used vulnerabilities, certain formerly well-liked varieties of malware seemed to be losing ground in 2022 [20]. Indications suggest that hackers are turning to discrete infections via email and the Internet of Things. Particularly in the case of ransomware infections, there is a persistent focus on big enterprises and governments as opposed to regular online users.

In 61% of the organizations in 2020, there was employee-to-employee malware activity. That percentage increased to 74% in 2021 and 75% in 2022. A total of 97% of firms experienced mobile risks, as noted by the security specialists at Check Point in the Mobile Security Report [21] in 2020. Additionally, 46% of businesses had a minimum of one individual who accessed a harmful mobile application. As of 7 August 2022, individuals attempting to access websites flagged as harmful by Safe Browsing saw almost four million browser warnings, according to Google’s Transparency Report [22]. The Statista Research Department published a report on 3 August 2022 [23] concerning the increased ransomware assaults which stated that there were about 236.1 million attacks globally in the first half of 2022. Five billion dollars USD in damage costs were reported in 2017, as well as eight billion dollars USD in 2018, eleven-and-a-half billion dollars USD in 2019, twenty billion dollars USD in 2021, and two-hundred-and-sixty-five billion dollars USD of damage costs by 2031 were reported and forecasted, respectively (Cybersecurity Ventures [24] for Ransomware). Some of the main effects of a malware assault include network failure, loss of important data, a rise in outgoing traffic, a slow computer, insufficient storage, unwelcome programs, DoS, DDoS, etc. Based on these reports, considering that malware created such chaos, finding quick and trustworthy methods to recognize malware and minimize their impact is urgently needed.

Over time, malicious software evolved and became more sophisticated. The use of anti-virus software is a part of the traditional method. The detection approach used by antivirus is frequently signature-based. A specific type of malicious software can be identified exceptionally well using signature-based identification, which searches for predetermined byte groups within an object. Since the new software signatures are not kept in the database, the biggest drawback is that it cannot detect malicious software that is zero-day in nature [25]. The anti-virus software, hence, is unable to obtain a fresh malware signature in time to detect new malware variants.

To fundamentally overcome the drawback of the signature-based approach, behavior-based recognition was developed. Instead of searching for the malware’s signature, it filters the framework’s behavior to identify any anomalous behaviors. The behavior-based process is limited by how long it takes the system to execute and how much more storage is needed. The program’s behavior during execution is the main emphasis of this approach. If a program runs normally, it is designated as benign; if not, malware is designated as the file. By dissecting the behavior-based method, we can particularly assume that its drawback is the production of a large number of misclassifications, given the ease with which a genuine application may be identified as malware or may run as a regular program.

Static analysis and dynamic analysis are the two main divisions used to classify malware analysis technology in the research community [26]. Malware can be rapidly identified using static analysis, but it can be concealed using packaging and other obfuscation techniques. Though it is susceptible to virus evasion, dynamic analysis successfully addressed the limitations of static analysis technologies. When solving the malware evasion issue in dynamic analysis, it is possible to circumnavigate software feature constraints by employing hardware features. It was demonstrated in [27] that malware may be effectively categorized by utilizing performance counters (such as IPC, cache behavior, memory behavior, etc.) to extract hardware attributes. In [28], hardware performance counters (HPC) were employed in conjunction with unsupervised techniques to detect malware based on the aforementioned studies. The classification accuracy of the aforementioned method, however, was not sufficient to accurately explain the behavior of malware using HPC.

The ability to deconstruct and analyze software in various ways is often heavily restricted in contemporary malicious software. Malware is often modified using obfuscation techniques, which involve altering the syntax of the code to make it more difficult to analyze while still preserving its functionality. Reverse engineering malware is more challenging when obfuscations are combined with the code optimization mechanisms often built into compilers. Obfuscations bring the task of malware classification into the scenario. The classification of malware is carried out using a variety of ML techniques, from models that need human feature engineering before training to DL models that can work directly with raw data.

Deep learning methods are inclined to overfit when they are trained on small-sized datasets, which is one of its disadvantages when compared to shallower models [29]. This can be an issue in disciplines such as program analysis, particularly in the categorization of malware, because it consumes much time and many resources for acquiring adequate instances with the actual information. Other industries, such as image identification and categorization, also frequently experience this issue [30]. The problem of not having enough training data can be readily solved in the domain of vision because new instances can be produced from previous knowledge by using some algorithmic modifications to the malware images, that do not change their semantics. This whole process is termed data augmentation and is an essential component of DL. It was first introduced in [31].

The visualization approach rose through the ranks of malware detection. Malware binaries present in the datasets or extracted from the opcodes are converted into grayscale or color images for examination and detection. Different types of image processing techniques were used for feature extraction and maintaining image uniformity. The combination of deep learning algorithms and visualization techniques was widely utilized and was proven to be beneficial. Deep learning models learn malware properties more accurately when trained on malware images. Deep learning models based on CNN are most utilized. Transfer learning [32] is yet another idea in malware analysis. It is the use of knowledge gained from accomplishing one activity to assist in the resolution of a separate, but related, challenge. In the context of machine learning (ML), this means that the layers’ learned parameters and weights from deep learning models trained over one collection can be applied to the analysis of some other collection. Transfer learning reduces training time by a significant amount for large datasets.

Transfer learning can be used to perform two types of tasks: classification and feature extraction. Pre-trained models in general are used mainly for two purposes, either for classification or feature extraction. ResNet34, ResNet50, VGG-16, ResNet101, and so on are a few examples. Pre-trained models are models that have already been trained to address a particular issue. As opposed to building a new model from scratch to address a similar problem, we start with the model that has been trained on another problem [33]. They are mostly CNN-based models.

The manuscript is structured as follows: Recent methods are elaborately discussed in Section 1.1, followed by the current study in Section 1.2. The proposed method and its mathematical expressions along with the datasets used are discussed in Section 2. The classification output results obtained by the proposed technique are specified in Section 3, following its detailed analysis through benchmarking in Section 4. Finally, the complete work is concluded in Section 5. 

### 1.1. Literature Survey

Marastoni et al. [34] presented a custom-trained transfer learning approach for malware detection. They used a custom OBF dataset prepared using obfuscation techniques for training the CNN model as a transfer-learned model. Another CNN and LSTM model were trained using the MalImg and MsM2015 dataset to verify prediction accuracy and used these models to predict each other’s malware dataset. They proposed the use of bicubic interpolation for image size uniformity and achieved on-par accuracy for LSTM. However, transfer learning demonstrated average accuracy without data balancing, the obfuscation techniques required improvement, and the models required fixed-size images, which increased the time needed for data pre-processing.

Casolare et al. [35] focused on the work of malware detection in Android applications. They employed a bespoke dataset of Android APK files from the Android malware repository, which were transformed into images, and then trained on well-known supervised machine-learning models such as J48, LMT, RF, RT, and REP Tree. The models achieved an accuracy of over 90% on the training data without using any benchmark datasets or data augmentation techniques. Their proposed approach identified whether the samples were malicious or benign, rather than detecting the specific malware classes. The difference in the datasets was based on the time difference during the creation of the obfuscated malware samples. This led to a decline in the model’s performance.

Kim et al. [36] proposed the use of CNN and MLP models for the analysis and detection of malware samples. They used the widely available Microsoft Malware dataset for their approach. To identify the group in which malware resides, portions of the dataset’s data were removed and transformed into an abstract visual graph. The degree of similarity between this malware was found using comparative analysis. Artificial intelligence deep learning was found to be an effective tool for quickly and accurately detecting malware through imaging. This method surpassed the traditional signature-based approach and could detect newly emerging malware. However, the MLP classifier was found to be unreliable and needed to be improved.

Khan et al. [25] proposed the use of EXE files for malware samples that were changed into opcodes and then into images for training and validation of two types of pre-trained models available—GoogleNet and ResNet. Their approach could convert the vindictive code into images. A small change in the image showed that there could be a large impact on the classification. This method outperformed existing dynamic and static methods. However, the pre-trained models are complex and require a great deal of expertise. It took more execution time and had a large validation loss.

Dai et al. [26] employed a hardware feature method in which a storage dump file’s contents were retrieved, turned to grayscale images, and then converted again into fixed-size images. Using the histogram of gradient, the image feature was retrieved. They used machine learning models such as MLP, KNN (K = 3 and 5), and Random Forest for malware classification. This method proved to be superior to dynamic methods and hardware features. The proposed system was unable to detect malware across the entire system and may have missed certain hardware components. The shallow models used resulted in sub-optimal accuracy without data balancing.

Singh et al. [37] proposed the method of converting the executables into image representation to eliminate the difficulties faced during static and dynamic analysis of the custom dataset created by collecting malware samples from the MalShare, VirusShare, and VirusTotal repositories. These data, along with images from the MalImg dataset, were utilized for validating the CNN and CNN-based ResNet-50 models. A visualization approach was used for the converted RGB images. The results were compared to the accuracy results of publicly available datasets. The performance was low for packed or previously unseen malware. The obfuscation technique evaded the visualization approach. The heavy-weight models used require extensive knowledge.

Venkatraman et al. [38] developed a brand-new, coherent hybrid neural network-based visualization technique. It described how to identify malware using image-based methods. The hybrid model was based on CNN BiLSTM and CNN BiGRU. The proposed models were validated on the BIG2015 and MalImg datasets. The performance was measured using various similarity measures. The scalability was noted by comparing it with available large malware datasets. This method employed similarity extraction, had a low processing cost, allowed for immediate training, and had comparable results. However, these were complex processing methods and require extensive knowledge of kernels and fine-tuning.

Vasan et al. [39] focused on the use of a novel ensemble CNN model for the efficient identification of packed as well as unpacked malware. They hypothesized that a deeper convolutional neural network (CNN) architecture would yield higher-quality features than traditional methods. They used an ensemble of ResNet-50 and VGG16 along with SVM on the MalImg and packed malware dataset obtained from VirusShare. Their method was resistant to obfuscation attacks, adapted to different datasets, could identify the benign samples, and reduced the count of misclassifications. However, constructing an ensemble model with complex and heavy-weight neural networks requires a deep understanding of the parameters to ensure compatibility with the other models in the ensemble.

Sharma et al. [40] employed a classification model for malware classification that made use of CNN and other machine learning classifiers in tandem with deep learning models to maximize their potential. With the aid of mathematical functions, these models enabled the detection of recently published malware. CNN alone performed best on the MalImg dataset in terms of method. The CNN-SVM model needed to be improved architecturally. This would involve using multiple SVMs for multi-class problems, which would increase the size of the model and, consequently, the computation time.

Naeem et al. [41] developed a novel classifier utilizing deep learning architecture based on CNN for multiclass classification. The malware binaries were converted into color images using a specific technique, which was then used to train a CNN tailored to ImageNet. Data augmentation was used during the fine-tuning process. It was observed that the RGB images outperformed grayscale images. They used the MalImg and IoT-Android malware datasets. Their approach was resilient to straightforward obfuscation techniques. The proposed CNN model was compared to ResNet50, VGG16, and InceptionV3 and was found to be more efficient, with lower computational costs, better evaluation of unbalanced data, and higher accuracy with colored images. It was also superior to other texture-based methods. However, this approach requires expert domain knowledge for fine-tuning through backpropagation.

Bakour and Unver 2021 [42] proposed a generic image-based classification approach for any file type (Manifest.xml, DEX, Manifest-ARSC-DEX, Manifest-Resources.arsc, Manifest-ARSC-DEX-Native_jar-based image dataset) that used grayscale images from the Android malware samples. For training classifiers such as RF, KNN, DT, Bagging, AdaBoost, and others, the local and global features were retrieved. Both the global and local characteristics were additionally classified using an ensemble voting classifier. They also compared the performance of their models with ResNet and InceptionV3 models. Their approach resulted in reduced computation time for individual classifiers and demonstrated that global features were more precise for malware categorization. The hybrid ensemble voting model was the most successful among all the models employed. Their approach followed the static analysis methods and could be affected by code obfuscation and manipulation. The model was vulnerable to injection attacks, and the ensemble model had a longer processing time. 

Kumar [43] suggested a refined CNN model called MCFT-CNN that could identify unidentified malware without the knowledge of feature engineering, binary code evaluation or reverse engineering, or even the most advanced dodging strategies. They employed transfer-learned fine-tuned CNN (trained on ImageNet) for the identification of malware present in the MalImg and BIG2015 datasets. The ResNet-50 model was used for comparison purposes. This approach was superior to traditional and existing transfer learning methods due to its low prediction time for unseen malware, high generalizability, and lack of need for feature engineering. Data augmentation was not taken into consideration here. This approach necessitates specialized knowledge of the domain and the models used necessitate uniform image size as input for categorization. 

Anandhi et al. [44] transformed the malware into Markov images along with the use of a Gabor filter to preserve the semantic information stored in them. The VGG3 and finely-tuned DenseNet201 models were then developed and trained on MalImg, BIG2015, and some benign samples that were collected, using this approach. The combination of Markov images and the Gabor filter proved to be more accurate for malware detection with less detection and execution time. The DenseNet201 model is a heavy-weight model with 201 layers and both models require uniform image size as input. Data augmentation could have been used in this approach for testing its effect on accuracy.

Pant et al. [45] focused on detecting the malware in grayscale image form. The VGG16 model, which was pre-trained, was used for transfer learning. Custom CNN achieved higher accuracy than VGG16, ResNet-18, and InceptionV3 when tested on the MalImg dataset. The custom CNN model outperformed the pre-trained models due to its ability to process non-uniform and limited data for malware classification.

Kumar et al. [46] leveraged the deep CNN architecture previously trained on ImageNet for classification. A CNN model was trained using grayscale images generated from Windows Portable Executable files as input. Early stopping was employed to prevent overfitting. The models were validated on two benchmark datasets—MalImg and BIG2015. The results obtained were comparable with other pre-trained models (VGG16, VGG19, ResNet50, and InceptionV3). The custom CNN was resilient to packed and encrypted malware. The models used were heavy-weight and the CNN was not trained on domain data. Fine-tuning the models proved to be challenging, and data balancing was not taken into account.

Kalash et al. [47] presented a general DCNN architecture for identifying malware. This method did not use manually created feature descriptors; instead, it learned the distinguishing representation from the data itself. The experiments were conducted on the MalImg and Microsoft benchmark datasets. When compared with the winning answer for the Microsoft Kaggle challenge, it incorrectly classified only three malware samples rather than fifteen. Though misclassification was less, it could not be compared to the winner’s solution as it used only ‘.bytes’ files instead of both ‘.bytes’ and ‘.asm’ files. The GIST + SVM approach was not very effective and needed improvement.

Unver and Bakour 2020 [48] generated three grayscale image datasets based on the file types (Manifest file-based image dataset, DEX code-based image dataset, Manifest-DEX-ARSC image dataset) which contained both malware and benign samples. They extracted both the local and global features and trained six different machine learning classifiers including RF, KNN, DT, Bagging, and Gradient Boost. They proposed a generic method that could be used for any type of app when converted into images. This approach demonstrated superior accuracy and faster computation time compared to previous dynamic and static analysis methods. The AdaBoost model turned out to be superior among all the classifiers. Their approach used static analysis techniques, which could be circumvented by code obfuscation and manipulation. It was not secure against injection attacks.

Jin et al. [49] divided the gathered malware data into various malware classes, then transformed them into image-type data. By retrieving the local features, they created an algorithm to uniformize training image size. Their proposed design was a set of autoencoders that incorporated CNNs for malware detection within their internal layers. They experimented on the dataset obtained from the Andro-Dumpsys study carried out by a Korean university to demonstrate the feasibility of their method. Autoencoders used an unsupervised detection method by observing the error value in malware reconstruction. With only a limited amount of data, the model was able to identify the relationship between malware data. It demonstrated superior performance and was able to withstand attempts to manipulate it. There was a separate encoder for each malware which increased the complexity and redundancy and decreased its scalability. It identified uncollected malware as benign and requires more resources and time.

Bakour and Unver 2021 [50] suggested a unique model in the field of malware classification called as DeepVisDroid, which focused on fusing deep learning techniques with image-based attributes. Four datasets of grayscale images were created by extracting malicious and benign datasets from the APK archives. DeepVisDroid was a 1D CNN-based model trained on local and global features. They also proposed 2D CNN and CNN inspired by VGG16 models. ResNet and Inception-V3 models were used for comparison purposes. Less computation time was observed for the global features-based DeepVisDroid model. DeepVisDroid based on ORB and KAZE features had less computation time than the other two features—SIFT and SURF. When compared to other models, DeepVisDroid proved to be more accurate and cost-effective. However, it was unable to detect obfuscation and code camouflage techniques such as injection attacks.

Lo et al. [51] employed a novel technique utilizing deep CNN and the Xception model to categorize the malware types. The Xception model was pre-trained on ImageNet, indicating that transfer learning was employed for the classification task. The evaluation was conducted using the MalImg and Microsoft malware datasets. This method was more flexible and effective in adapting the malware evolution. The Xception model took less time and did not require domain expertise. This reduced the problem of overfitting and produced similar results. To optimize these heavy-weight models, a thorough understanding of them is necessary. Additionally, knowledge of the conversion process of ‘bytes’ and ‘asm’ files into images is necessary. Data augmentation was not taken into consideration.

Parihar et al. [52] proposed the S-DCNN model to tackle the malware detection problem by utilizing the ideas of ensemble learning and TL. The S-DCNN model comprised three CNN models, namely Xception, ResNet50, and EfficientNet-B4. The heterogeneous information of each intermediate model was combined through the ensemble technique, producing models with high generalizability and low variance. It did away with the older usage of reverse and feature engineering, deconstruction, as well as other domain-specific techniques. The model was tested using MalImg [53] and data obtained from VirusShare. The ensemble model contained three highly heavy-weight pre-trained models and data augmentation was not taken into consideration. Such models could be complex to handle and understand.

Darem et al. [54] focused on the work of malware classification or identification by using the obfuscated malware opcodes present as ASM files that were converted into grayscale images during the processing stage. The images were generated based on the features extracted during the feature engineering stage. Opcode, segment, and pixel count were the three primary features, whereas the number of lines and characters were the secondary features that were extracted and analyzed with the help of Random Forest. An ensemble semi-supervised approach combining CNN and XGBoost models was used for the classification purpose. The proposed model fared well with a 99.12% accuracy score when compared with other approaches. In this approach, no such benchmark dataset was utilized and it required extensive pre-processing tasks and knowledge. A small dataset containing only nine types of malware was used in this approach. 

Roseline et al. [55] employed an ensemble Deep Forest algorithm which was similar to deep learning techniques along with the vision-based approach. The model used fewer hyper-parameters and, hence, did not require too much tuning. Different types of forests and random sampling were carried out for the high dimensional data. The ensemble model was used along with three benchmark datasets—MalImg, BIG2015, and Malevis. Their model demonstrated superior generalizability, accuracy, precision, and reduced computational overhead. The proposed approach learned patterns from the data itself, rather than relying on specially extracted features, resulting in a simpler model, and avoiding overfitting. Multiple experiments were conducted based on different forest combinations. The efficiency of the approach was tested by classifying an unseen benchmark dataset, Malicia. It surpassed other deep learning and shallow machine learning models in terms of accuracy, precision, and f1-score. Data augmentation was not considered while the experiments were conducted.

Ding et al. [56] used the bytecode image of malware APKs to focus on the work of Android malware detection. They employed a general CNN model for classification purposes. Multiple experiments were conducted based on the number of convolutional layers present in the CNN model. They suggested utilizing simpler data processing and categorization methods, as well as creating higher-order feature maps based on CNN maps to enhance the learning capacity of the models. For their purpose, they used a malware dataset provided by the DRE-BIN project conducted by the University of Gottingen along with 1000 benign samples. The malware dataset used was of small size, with only 14 malware families. This approach was able to detect encrypted malware that evaded traditional detection methods, without the use of a benchmark dataset or data augmentation. They achieved a satisfactory accuracy score but did not outperform other existing deep learning approaches.

Ngo et al. [57] performed a complete experimentation on the existing approaches based on static analysis for IoT malware detection. The static characteristics included CFGs, strings, opcodes, grayscale images, file headers, etc. For the grayscale image-based approach, they employed a CNN model. For other characteristics, shallow ML models such as KNN, DT, SVM, RF, etc. were used. All the experiments were conducted on an IoT malware dataset provided by the IoTPOT team and VirusShare. Some benign samples obtained from IoT SOHO were combined along with the malware dataset. The ELF-header, String-based, Opcode-based, and PSI-graph approaches fared well when compared to the grayscale image-based approach. The accuracy of the image-based approach was reduced when obfuscation or encryption techniques were applied. Despite this, it demonstrated impressive scalability due to its ability to protect against node failures and its straightforward design. In this approach, the benchmark dataset was not utilized for any of the experiments.

Huang et al. [58] focused on the work of detecting malware present in Windows OS. They proposed a hybrid visualization approach along with deep learning models for malware classification. Both static and dynamic analysis methods were used to obtain the hybrid technique. In this approach, they utilized a Cuckoo Sandbox to perform the dynamic analysis. If the dynamic visualization was for malware, the hybrid visualization would also not work. In the visualization process, they obtained RGB images instead of grayscale images. No such benchmark dataset was used. A collection of malware and benign samples obtained from ‘virussign.com’ was obtained. The hybrid images were trained on a VGG16-based neural network model. The model trained on hybrid images outperformed the model trained only on static visualized images. Overall, the proposed approach outperformed a few of the existing approaches by achieving a satisfactory accuracy score. The proposed model was not trained sufficiently on old malware types and required a continuous update to increase its ability to identify unknown malware. The whole implementation was carried out on a VM with limited installed applications; thus, the actual behavior of the malware could be found during the dynamic analysis method.

Naeem et al. [59] proposed a hybrid architecture that combined the knowledge of image visualization and deep learning models for malware detection in the field of industrial IoT. This research focused on the utilization of color images. To evaluate their approach, two datasets were employed: the Leopard Mobile dataset and the MalImg dataset. These datasets were used to compare the results with other methods. Initially, the APK files were converted into color images during the processing stage. Then, the images were directly sent as input to deep CNN models for training and classification. The experiments were conducted for two different image dimensions, i.e., 224 × 224 and 229 × 229. The proposed model achieved better and higher accuracy for the benchmark MalImg dataset when compared to the Leopard Mobile dataset. The images with dimensions 229 × 229 returned better results when compared to the images with 224 × 224 dimensions. This approach was widely adopted by many authors recently and no data balancing was conducted during the data pre-processing stage. The classification time for the proposed method was longer than other algorithms that had been used before.

He et al. [60] also focused on CNN-based models to classify malware by using image representations. The primary objective of the study was to assess the efficacy of the proposed model in combating superfluous API injections. To this end, they proposed the use of spatial pyramid pooling (SPP) layers to address the issue of varying input image sizes. The notion was to reduce the data loss because of fixed input image size. They used both color and grayscale images in their implementation. The dataset used was obtained from the Andro-Dumpsys study conducted by Korea University. The results obtained implied that RGB images worked better with the ResNet model, whereas the grayscale images worked better with the simple CNN model. RGB images were not as effective as grayscale images when it came to detecting redundant API injection. Additionally, the use of the SPP layer in the classification task was not successful due to memory limitations. The models used were not optimized properly to achieve the author’s goal. Redundant instructions were injected at the binary level as compared to the source code level because of dataset constraints which might have been altered by the compiler, resulting in a different type of outcome.

Su et al. [61] proposed a CNN-based approach to mitigate the risks of DDoS attacks in the IoT environment. They used a two-layered light-weight CNN model to classify the malware dataset which was prepared by using benign samples collected from their Ubuntu’s System files and malware samples from the IoTPOT dataset. In the pre-processing stage, the malware binaries were converted into image representation and data balancing was carried out. Two different types of experiments were conducted—two-class classification, where samples were classified as benign or malware, and another three-class classification, where classification was carried out between two malware classes and a benign class. The two-class classification yielded a better result than the three-class classification experiment. No advanced image processing techniques or complex models were employed in this model, which made it susceptible to obfuscation techniques.

Asam et al. [62] designed a CNN-based malware detection architecture in IoT called iMDA. Edge exploration and smoothing, multi-path expanded convolutional operations, and channel compressing and boosting in CNN were just a few of the feature learning schemes that were included in the proposed iMDA’s modular design and were used to learn a variety of features. Edge and smoothing operations that were implemented in the split-transform-merge (STM) blocked learn the variations in the malware classes. The multi-path expanded convolutional procedure was employed to identify the overall malware pattern structure. Concurrently, channel compressing and merging assisted in controlling complexity and obtaining a variety of feature maps. Data augmentation was carried out in the initial processing stage. On a benchmark IoT Malware dataset, the proposed iMDA’s performance was assessed and contrasted with several state-of-the-art CNN architectures such as AlexNet, VGG16, ResNet50, Xception, GoogleNet, etc. The proposed model demonstrated superior performance compared to existing models when applied to the IoT Malware dataset. Its impressive ability to distinguish between malicious and benign files suggested that it could be extended to include both IoT Elf files and malware detection for Android-based devices.

Makandar and Patrot [63] used machine learning models such as SVM and KNN (K = 3) for malware classification. The Gabor Wavelet, GIST, Discrete wavelet Transform, and other features were employed to construct an efficient texture feature vector using multi-resolution and wavelets. These feature vectors were used to train the models. Feature selection was carried out using Principal Component Analysis (PCA). The Malheur and MalImg datasets were used for classification purposes. The SVM model achieved an average accuracy of 98.88% over all the malware families, whereas the KNN (K = 3) model achieved an average accuracy of 98.84%. The KNN model achieved the best results with the Euclidian distance metrics. Overall, the SVM model was better than the KNN approach. The classification error obtained was much less compared to the existing methods. Data balancing was not performed in this approach. Table 1 presents a comprehensive overview of the literature review, including information on the techniques employed, datasets used, objectives, and limitations.

### 1.2. Current Study

For malware classification, features extracted from pre-trained models were trained and categorized using models such as Logistic Regression, SVM, Random Forest, Decision Tree, and so on. Although pre-trained models offer the advantage of faster training and good accuracy, they also have certain drawbacks. Pre-trained models are heavy-weight complex models that require more resources and knowledge to implement. It can be challenging to adjust the parameters of these models to suit our requirements, as they have not been trained on data from our specific domain.

Work carried out in the field of malware detection involves excessive dependency on widely available pre-trained models such as GoogleNet, ResNet, VGG, Xception, and Inception. The most-used neural network model is CNN. Shallow machine learning models which are directly fed with features for classification include SVM, KNN, Random Forest, Decision Tree, Bagging, AdaBoost, Gradient Boost, ensemble voting classifier, LMT, J48, etc. The most general procedure involves the conversion of executable files such as Windows PE files, Android APKs, bytes, .asm, etc., into images and then the application of image recognition using deep learning algorithms. The use of pre-trained models brings the concept of transfer learning into this scenario. Comparative work between pre-trained and simple machine learning models such as KNN [64] has been carried out already. The benchmark datasets that were used in this field include Microsoft BIG, MalImg, and MsM2015. Research was also conducted on an IoT Android mobile malware dataset to monitor malicious activity on mobile devices [41].

In this paper, two malware image datasets—Malevis [65] and MalImg [53]—were used. Both malware image datasets underwent pre-processing, and each image was resized to 224 × 224. Since both datasets were small, the oversampling technique was used to increase the data. The MalImg dataset served as the baseline for evaluating the precision of our proposed model. The Malevis dataset was used to build a CNN-based model for transfer learning. The features of the images in the MalImg dataset were extracted using this CNN-based model. Utilizing a transfer learning model that was developed from scratch is what this method entailed. The proposed model was applied to the retrieved features, which consisted of three neural network architectures that were not commonly used for malware classification: the autoencoder, the gated recurrent unit, and the multi-layer perceptron. The autoencoder model encodes the extracted features of each malware image into a sequence of features that are equivalent to the types of malware present in the dataset. To evaluate the viability of the GRU model for malware detection when employed separately, the feature sequence obtained as output from the autoencoder model for each instance is re-generated using the GRU model. The MLP model performs the final classification by taking the output from the GRU model as its input. Through an examination of the classification errors, the strengths and drawbacks of the trained models were brought to light.

The major contributions of this work are:Transfer learned model trained from scratch on domain data;The design of a light-weight system for malware classification which consumes less time and resources;No specialized domain expertise is necessary for understanding and fine-tuning the model;Comprehensive comparison of methods;Effect of the use of data augmentation in malware detection;Use of a benchmark malware dataset to validate the model’s

Validation of least used DL models in the task of malware classification.

## 2. Materials and Methods

Numerous proposed systems were used to identify malware, and these systems used a variety of data processing techniques, feature extraction techniques, transfer learning techniques, and algorithms. Previous studies used time and resource-intensive complicated feature extraction and data processing techniques, which many attempted to make up for by employing freely accessible pre-trained classification models or using shallow machine learning models. These heavy-weight pre-trained models, which require expert expertise to fine-tune the parameters, were typically learned on big image datasets that did not correspond to the malware domain. Hence, in this paper, a straightforward data processing method using Python 3.10.1 released by Python Software Foundation located in Wilmington, DE, USA and a compact, basic, and easily adjustable ensemble neural network model that leverages transfer learning for feature extraction was proposed. Figure 1 depicts our suggested system’s fundamental architecture or workflow.

### 2.1. Datasets Used

This section details the malware datasets that were utilized in our work. Two datasets were used—MalImg and Malevis datasets. The Malevis dataset that was used for transfer learning is discussed below, followed by the benchmark MalImg dataset.

#### 2.1.1. Malevis

The Multimedia Information Lab of Hacettepe University’s Department of Computer Engineering, in partnership with COMODO Inc., Clifton, NJ, USA, compiled the open-source Malevis dataset. There are byte pictures from 26 different classes in this corpus. First, using the bin2png script, which was created by Sultanik, binary images in 3-channel RGB form were extracted from malware files (provided by COMODO Inc.) to create this corpus. The vertically long images were then resized into 2 square dimensions (224 × 224 and 300 × 300 pixels). This dataset is accessible in [66] as well as in Kaggle headquartered at San Francisco, CA, USA. 

The total RGB images in the Malevis dataset were 9100 training and 5126 validation images. While each class in the testing dataset held a different number of images, all the training classes had 350 image samples. The dataset’s directory structure was created so that it could be used later without additional work. In this sense, we could use it in a variety of deep learning frameworks, including Caffe, PyTorch, Tensorflow, and Keras. This dataset was utilized for training the CNN-based transfer learning model for the sole purpose of feature extraction.

#### 2.1.2. MalImg

The benchmark MalImg dataset that was used in many existing works in the field of image-based malware detection contains around 9458 malware instances distributed among 25 different classes. This dataset contains the malware instances directly in the form of grayscale images instead of their corresponding malware programs. This reduced the time duration of the data processing stage by a significant amount. The executable file bytes were quickly converted to floating-point numbers, which were then used to represent the image’s grayscale pixel values. A severe imbalance existed between the categories in the dataset. The largest class, (‘Allaple.A’), had 2949 instances, while (‘Skintrim.N’), had only 80 instances.

In Figure 2, 5 malware samples of 5 different classes are shown. Images belonging to different classes possessed distinct characteristics that made it easier to distinguish between them. This factor drove the work in [67]. However, there were also a few classes that had minimal differences between their patterns making it difficult to differentiate. In Figure 3, we observed that the images of the two classes ‘Swizzor.gen!E’ and ‘Swizzor.gen!I’ were very similar to each other. Classifying them 100% accurately was a problem even after using many existing systems.

### 2.2. System Requirements

The complete implementation of the proposed model for malware detection was carried out on a system which had Windows 11 64-bit OS with 16 GB RAM and Python 3.10.1 released by Python Software Foundation located in Wilmington, DE, USA, installed. The widely used and open-source Visual Studio Code was used for implementation purposes. The libraries Numpy 1.23.4, TensorFlow 2.10.0, Pandas 1.5.1, PIL 9.2.0, and Sklearn 1.1.2 were installed and used along with the Jupyter extension in it.

### 2.3. Data Pre-Processing

Bicubic interpolation [34], conversion of APK files, .EXE files, and malware binaries into images, conversion of the dataset into abstract visual graph [36], using hardware features [26], visualizing malware as Markov images along with Gabor filter [44], etc., are some of the widely used image pre-processing techniques that have been used in the field of malware detection. These methods require precision, time, resources, and a good amount of knowledge before carrying them out. To conserve time and resources, employing straightforward methods of altering the size, shape, and format of our image data was proposed.

Both datasets contained images of malware which saved us the process of conversion of executable malware files into images. To assess the impact of data balancing on malware identification and classification, the oversampling technique for data augmentation was employed. This was accomplished without the use of any Python programming techniques. A total of 73,725 images belonging to 25 malware classes were obtained for the MalImg dataset after data augmentation. For our purposes in this work, using images that were 224 × 224 in size was proposed. Since the Malevis dataset was available in two square dimensions, directly the 224 × 224 sized images were used, whereas the MalImg dataset contained images of different dimensions which were resized into 224 × 224, using the ‘resize ()’ of the Image library in Python.

Images from only one virus class at a time were processed and saved individually. Using the ImageOps library’s ‘grayscale ()’ function, the images in the Malevis dataset were transformed into grayscale images to achieve the efficiency of grayscale images in malware detection as compared to RGB images. The ‘asarray ()’ function was then used to break down the images into their pixel data. The resulting array, which contained the pixel values of each image, was first reshaped into a 3D array of dimensions (224,224,1) so that they could be used as input to the models. The ‘1′ here represents only 1 channel for grayscale images. (If RGB images had been used then the value 3 would have been used). Next, they were converted into float data type for normalization, and finally, they were saved as “.npy” array files.
Normalization function = (Pixel Value)/(255.0)(1)

### 2.4. Transfer Learning

Malware detection made substantial use of the concept of transfer learning. By employing the readily accessible pre-trained models such as VGG16 [39,41,45,46], Inception-V3 [46], Xception [51,52,62], ResNet [24,37,39,43,46,52], and others, existing works exploited transfer learning in the form of feature extraction or classification. These pre-trained models were typically developed using sizable image datasets outside the domain of our actual work. Although these models produced positive results, it can occasionally be challenging and complex to fine-tune these models to meet our goals for malware detection. Following a similar approach to [34], employing a CNN model for transfer learning trained from scratch on the Malevis dataset was proposed. 

The pre-processed data of the images in the Malevis dataset were used for training a CNN model. The CNN-based model was built with only 12 layers, which were much fewer than other pre-trained models such as ResNet-50, ResNet-34, ResNet-101, DenseNet-201, Xception, etc., considering the goal of building a light-weight model. As a result, training can take less time and use fewer resources. The CNN model contained 3 convolutional 2D layers, each of which was followed by two other layers, namely Max Pooling 2D and Dropout layers. It then had two dense layers immediately following a Flatten layer. The model was sequential.

The convolution kernel of the convolutional 2D layer, which was combined with the layer’s input, yielded a tensor of outputs. When the convolutional layer was used as the initial layer, the parameter ‘input_shape’ was set as a combination of integers or none, but without mentioning the sample axis; for example, ‘input_shape’ = (224, 224, 1) for 224 × 224 grayscale images used in our case. The number of filters, kernel size, and activation function was set as 32, (3,3), and ‘relu’, respectively, for each of the convolutional 2D layers with an additional ‘input_shape’ attribute set for the first layer only. The MaxPooling 2D layer used each input channel’s maximum value throughout an intake window of a size given by ‘pool_size’ to downsample the intake along its axes (height and breadth). A dropout layer during the training period randomly set input units to 0 at each step to avoid overfitting. Using a Flattening layer, the resulting 2D arrays of pooled feature maps were compressed into a unique long continuous linear vector. A dense layer’s neurons were given one output from the layer above, which was received by each neuron in that layer. The dense layer applied the formula:Output = activation (dot (input, kernel) + bias)(2)
where the following parameters describe the dense quality: Activation—refers to the activation function for each element;Kernel—refers to a weight matrix that was generated by the layer;Bias—is a layer-generated bias vector (applicable if using bias = True).

Figure 4 shows the detailed architecture and parameter values of our model. The architecture of the model was plotted using Model Plotting Utilities [68] and Graphviz [69]. Only the necessary parameters were specified as values in each layer, with the remaining parameters set to their default values. In comparison to the default parameters, the key parameters were simpler to comprehend and alter. The 26 neurons in the last dense layer corresponded to the count of malware families present in the Malevis dataset. Because the Malevis dataset had multiple categories, the loss parameter for the model was set to be “sparse categorical cross-entropy”. The accuracy measures were chosen to show the training and testing accuracy and loss values for each epoch.

The CNN model trained on the Malevis dataset was used for feature extraction of the malware instances present in the MalImg dataset. The last dense layer with 26 neurons was removed, and the trainable parameter was set to False for all other layers. A second compilation of the model was performed. For extracting the features from the malware images, the pre-processed MalImg dataset was fed to this recently assembled model. Each malware was represented as a sequence of 64 features, as the output layer of the trained CNN model had 64 values. To extort the features from the images in the MalImg dataset, the model employed previously learned weights from the Malevis malware dataset. The weights were based on malware images instead of some random images, ensuring more accuracy in the feature extraction process. This method of extracting features from malware images in the MalImg dataset using a pre-trained CNN model represents the idea of transfer learning for feature extraction.

### 2.5. Ensemble Model

Our proposed ensemble approach comprised three neural network models, namely autoencoder, gated recurrent unit (GRU), and multi-layer perceptron (MLP). Such a combination of NN was not previously utilized in the domain of malware detection. Previous works included extensive usage of CNN, CNN-based pre-trained models [25,37,39,41,43,44,45,46,50,52,62], shallow models such as Logistic Regression, Random Forest, Decision Tree, Bagging, SVM, etc., or an ensemble of these shallow models [35,42,48,57].

#### 2.5.1. Autoencoder

Feedforward neural networks called autoencoders have inputs and outputs that are identical [70]. It is an unsupervised learning technique. From this representation, they recreate the output after compressing the input into a reduced code. The code often referred to as the latent-space representation is an “abridged” or “compressed” version of the input. An autoencoder is made up of three components: first, an encoder is followed by a code which in turn is followed by a decoder. The encoder creates the code by compressing the inputs, and the decoder then uses that code to reconstruct the input. The architecture of the decoder and the encoder are mirror images of each other. The 4 hyper-parameters that need to be handled before we start training the autoencoders are code size, no. of layers, no. of nodes per layer, and loss function. Code size refers to the node count in the middle layer. Since the layers are piled one on top of the other, the autoencoder design that was proposed is known as a stacked autoencoder. With each additional layer of the encoder, there are fewer nodes per layer, which increases in a similar order in the decoder. 

Jin et al. [49] employed the use of a collection of CNN-based autoencoders for malware detection. They observed the error value in malware reconstruction to classify the samples. They used separate encoders for each malware, thereby increasing the complexity of the model. Our approach was distinct from theirs as there was no attempt to categorize malicious software using autoencoders. Our aim was to reduce the size of the extracted feature sequence so that it contained all the necessary information. Dimensionality reduction leads to smaller datasets, which require less time and resources to train. Therefore, what is required is a model with intermediate coding levels and an encoder. However, first, to determine the encoding effectiveness, a complete autoencoder is needed.

Figure 5 shows the architecture of our proposed autoencoder model. The encoding and decoding layers of the model were made up of dense, batch normalization, and LeakyReLU layers instead of CNN, as used in [49]. This ensured that the training time was reduced much more when compared to CNN. The code size was determined to be the same as the count of classes in the MalImg dataset, i.e., 25, and a dense layer was used to create the middle layer. The model was configured with an input size of (64,) units and an output size of 64 units, where 64 represents the length of the extracted feature sequence. The general architecture of our autoencoder model consisted of 2 encodings, 1 bottleneck or middle code layer, and 2 decoding layers. The loss and activation functions were set as ‘mean squared error (mse)’ and ‘linear’, respectively. All other parameters in each layer were set to their default values. During model compilation, the ‘metrics’ parameter was set to ‘accuracy’ to view the accuracy scores and loss values. After training the autoencoder with the MalImg feature dataset, a new model named Encoder was extracted from it, which comprised the trained encoding and middle code layers. Then, the compiled Encoder model was used to encode the MalImg features into a 25-unit long sequence. The encoded features were sent to the gated recurrent unit (GRU) model for further processing.

#### 2.5.2. Gated Recurrent Unit

Cho et al. presented the GRU [71] to handle the vanishing gradient concern that emerged in a typical RNN. It is a different kind of LSTM model. For the gradient problem, an update and reset gate was used. These two vectors became factors for determining what should be output. They can be educated to remember data accumulated in the past without letting it fade out over time or to dismiss details that are irrelevant to the prediction.

The model used the update gate to determine whether it could duplicate the data accumulated earlier and, therefore, completely avoid the vanishing gradient concern [47]. This was found out using the formula:z_t_ = σ(W^(z)^x_t_ + U^(z)^h_t−1_)(3)

When x_t_ is connected to a network entity, it is multiplied by its weight W^(z)^. Similarly, multiply by U^(z)^ for h_t−1_, which contains t − 1 units of data in the past. To produce an outcome in the range of (0,1), a sigmoid function is used over the sum of these two values.

The reset gate helps to decide what amount of prior data needs to be retained. The formula to calculate that is similar to the update gate formula [72]. The weights and how the gate was used made a difference.
r_t_ = σ(W^(r)^x_t_ + U^(r)^h_t−1_)(4)

The reset gate was used to create new memory content that would retain relevant historical information and was computed by:h_t_′ = tanh (Wx_t_ + r_t_ ⊙ Uh_t−1_),(5)
where h_t−1_ multiplied by its weight U is element-wise multiplied with r_t_ and then added to x_t_ multiplied by its weight W. Finally, tanh is used to obtain the result.

The network computes the h_t_ vector, which contains information about the current unit and is transmitted to the network via the update gate. To identify what needs to be gathered, it looks at the contents of the present memory (h_t_′) and h_t−1_.
h_t_ = z_t_ ⊙ h_t−1_ + (1 − z_t_) ⊙ h_t_′(6)

In this field of malware detection, GRU is one of the least frequently employed neural network models, yielding average accuracy values. On benchmark datasets, Venkatraman et al. [38] classified malware using a CNN-based BiGRU (bidirectional GRU) model. For analyzing the performance, they employed similarity measures. These models call for a deep understanding of kernels, layers, and parameters for fine-tuning. In more recent work using GRU for malware identification [73], the behavioral characteristics of CPU, disk consumption, and memory of programs running in cloud-based platforms without restriction were collected to categorize malicious apps. The primary reason for incorporating GRU into our proposed ensemble model was its shorter training time, which reduced the overall training time of the ensemble model. It was also planned to assess its effectiveness in detecting malware if it was supplied with enhanced malware characteristics instead of raw features. A many-to-many prediction GRU model based on the malware labels was implemented. The GRU model can take the encoded features obtained from the Encoder model and use them for educating the model along with the malware classes. The trained model can be used to predict the sequences learned, and to determine its efficiency in learning the relation and differences between various malware classes. Figure 6 shows the proposed GRU architecture and its working. No specialized layers were utilized. GRU layers with varying units and default parameter values were used. The ‘return_sequences’ parameter was set to True, to forward the sequence to the following GRU layer. For each GRU layer, the activation parameter was set as ‘tanh’ (as implied in the above formulas). The dense layer was the final layer with 25 units that gave the resultant sequence. The ‘softmax’ activation function was used. During compilation, the loss function was set as ‘sparse_categorical_crossentropy’ because of multi-class classification, and the ‘metrics’ parameter was set as ‘accuracy’.

The GRU-predicted sequences were then forwarded to the next model in the ensemble architecture for final classification, namely the MLP model.

#### 2.5.3. Multi-Layer Perceptron

The next and final model in the ensemble approach was the multi-layer perceptron or MLP model. The MLPClassifier model available in the sklearn library was used. MLPClassifier evaluates the partial derivatives of the loss function of the parameters at each time interval as part of its iterative training procedure to adjust the parameters. To avoid overfitting, the loss function might additionally have a regularization term added that reduces the model parameters. Floating point data arrays in the form of dense NumPy or sparse scipy arrays are supported by this implementation [74].

The MLP classifier employed in [36] outperformed the traditional malware detection technique; however, it was found to be unreliable and in need of improvement. In [26], the MLP classifier was found to be more effective than dynamic methods and hardware features, but its accuracy was unsatisfactory without data balancing. One of our goals was to assess the effectiveness of models when data augmentation is used, thus eliminating one potential cause of low accuracy. The idea is to achieve a better and more stable MLP classifier for malware detection when refined and restructured features are provided for training. The previous two models—autoencoder and GRU—refined the information contained in the features and the MLP classifier was used to verify if these changes had a positive response on the classification accuracy.

The MLP classifier used contains 3 hidden layers, each with 100 neurons. The activation function, max iterations, and solver were set as ‘relu’, 1000, and ‘adam’, respectively. Except for the ‘max_iterations’ and ‘random_state’ parameters, all the parameters were set with default values. In terms of learning time as well as validation score, the default solver ‘adam’ performed admirably on somewhat sizable datasets (with tens of thousands or more training samples). When solver = ‘sgd’ or ‘adam’ was used, batch sampling and train-test splitting were determined by the ‘random state’ parameter, as well as the creation of random numbers for initializing bias and weights. In our case, the ‘random_state’ parameter was randomly chosen as 42. The proposed MLP classifier is a very basic model. This will help in creating a model that is light-weight, straightforward, and easy to comprehend.

## 3. Results

The efficiency of the SE-AGM model is discussed in this segment. Two types of experiments were carried out for the proposed model. Experiment 1 used the extracted features of the MalImg dataset in an 80:20 ratio for training and testing data, whereas Experiment 2 used the 70:30 ratio for the same purpose. Within the ensemble model, the autoencoder model and GRU model were trained for 10 epochs and 500 epochs, respectively, with a default batch size of 32. The MLP classifier with three hidden layers each of 100 neurons was trained for 1000 iterations, as mentioned earlier in Section 2.5.3. In each experiment, the MLP training and classification were carried out five times to obtain an average accuracy value.

To evaluate our SE-AGM model’s effectiveness in classification, some metrics were required. In this paper, four metrics were calculated to give an insight into our model, namely accuracy, recall, f1-score, and precision. These metrics were calculated based on four values—True Positive (TP), True Negative (TN), False Positive (FP), and False Negative (FN). TP signified a test outcome that accurately identified the existence of a condition or trait. TN signified a test outcome that accurately demonstrated the absence of a condition or trait. FP indicated an inaccurate test result that claimed the presence of a certain condition or quality, whereas FN indicated an inaccurate test result that claimed the absence of a certain ailment or quality.

Table 2 displays the average training as well as testing accuracy of SE-AGM for both Experiments 1 and 2 performed on the MalImg dataset. The accuracy values were rounded off to two decimal places for approximation. The average accuracy achieved in Experiment 1 for training and testing was 99.37% and 99.30%, respectively, while in Experiment 2, it was 99.45% and 99.43%, respectively. The results obtained demonstrate the effectiveness of our proposed model in both experiments on the MalImg dataset, thus validating our ensemble model approach for malware detection. When the train-to-test ratio was taken as 80:20 instead of 70:30, the proposed SE-AGM model performed slightly better. 

Table 3 and Table 4 show the average recall, precision, and f1-score computed for both Experiments 1 and 2 using the sklearn’s ‘classification_report()’ function [75]. From these tables, it is evident that our proposed ensemble model had difficulty in accurately identifying four malware classes, namely ‘Allaple.A’, ‘Allaple.L’, ‘C2LOP.gen!g’, and ‘C2LOP.P’, whereas it was 100% efficient in identifying the remaining 21 malware classes. The malware classes ‘Allaple,A’ and ‘Allaple.l’ were slight modifications of one another, as was the case for ‘C2LOP.gen!g’ and ‘C2LOP.P’. This suggests that identifying similar malware classes is challenging for the proposed model. In both the experiments, ‘C2LOP.gen!g’ and ‘C2LOP.P’ has the least F1-score. Maximization of this metric value would improve our proposed model’s performance.

It is evident from Table 3 and Table 4 that Experiment 2, when compared to Experiment 1, had equal or higher metrics value for the four classes ‘Allaple.A’, ‘Allaple.L’, ‘C2LOP.gen!g’, and ‘C2LOP.P’, thereby again implying that Experiment 2, with 80:20 train to test ratio, had a better performance with our proposed ensemble method.

## 4. Discussion

Our proposed model’s accuracy was compared with existing works that focused on the benchmark MalImg dataset such as IMCEC by Vasan et al. [39], S-DCNN by Lo et al. [51], and layered Random Forest ensemble by Roseline et al. [55]. Both IMCEC and S-DCNN employed ensemble techniques, similar to the one used in our malware classification system, albeit with slight differences in the models used in the ensemble approach. In contrast to S-DCNN, which employed a combination of ResNet-50, Xception, and EfficientNet-B4, IMCEC used an ensemble of pre-trained Xception models.

On comparing the efficiency of our SE-AGM model with existing ensemble approaches, it was observed that our model achieved comparable and equivalent accuracy values compared to existing works. The S-DCNN model used in [51] and Experiment 2 (80:20) with our SE-AGM model had the same accuracy value of 99.43% for the MalImg dataset. The IMCEC model proposed in [43] with an accuracy value of 99.50% was superior to S-DCNN and our proposed model by a small margin of 0.07%. Our model outperformed the Random Forest ensemble approach by a margin of 0.78%, indicating its superiority. Hence, our model stood second in the case of classification accuracy. Experiment 1 of our proposed model yielded the least accuracy of 99.30% among other models, indicating that more training data were needed to improve the accuracy of malware detection. This further reinforced the need for models to be trained with extensive malware datasets containing a wide range of classes to effectively detect modern malware attacks. Table 5 and Figure 7 show the comparison between the accuracy levels of our proposed model and other existing ensemble models.

Our proposed ensemble model, SE-AGM, had the advantage of achieving 100% recall, f1-score, and precision in identifying and distinguishing between malware classes ‘Swizzor.gen!E’ and ‘Swizzor.gen!I’, whereas other proposed models such as transfer learned CNN and LSTM model in [34], CNN and ResNet-50 in [37], IMCFN in [41], and MCFT-CNN in [43] struggled to achieve this. All these models had precision, recall, and f1-score values lower than 90% when attempting to identify the two malware classes mentioned above. Some models even had these metrics below 50%. Although our proposed model had some difficulty in distinguishing between certain classes, the metrics obtained were still comparable to those of existing methods. When compared in terms of accuracy (Table 6), our proposed ensemble approach outperformed all the other non-ensemble approaches in malware classification on the MalImg dataset; thus, making our model more reliable than others.

Further investigation could involve training the model with a larger dataset that encompasses a range of malware categories to enable the recognition of different types of malwares. To address the rapid growth in malware attacks, enhancements could be made to enable real-time training and malware detection. To confirm the efficacy of the suggested transfer learning-based ensemble approach, this study could also be evaluated on other benchmark malware image datasets. Researchers could benefit from using this methodology to create a more dependable security solution.

The proposed SE-AGM model has certain drawbacks, such as its incapability of handling malware images of different sizes and recognizing obfuscated malware. Additionally, the classification process was prolonged due to the need for pre-processing the unknown malware image before classification. If the model encountered a malware sample that it had not been trained on, it would be difficult to determine whether it was a benign sample or a different type of malware. This challenge is also present in current research in this field, as well as the inability to effectively handle real-time encounters with disguised malware. This fact is the driving force behind the increased study in this field.

## 5. Conclusions

An ensemble of neural network models, SE-AGM, which consisted of an autoencoder, a GRU, and an MLP, was proposed, with the output of one model serving as the input for the next. During the data pre-processing stage, no special techniques were utilized. During the pre-processing stage, important tasks such as data augmentation, image resizing and reshaping, conversion of an image into pixel data, and normalization were carried out. The ensemble model was trained with malware features obtained from the MalImg dataset extracted using the concept of transfer learning for malware classification. For transfer learning, a CNN model was educated using another malware image dataset, the Malevis dataset. The last dense layer in the trained CNN model was removed and the new model was again compiled. This new model was then used for feature extraction. The autoencoder, an unsupervised model, was used to encode the long feature sequences into short sequences retaining the essential information about the malware. The GRU model was chosen because of its capability to develop with less training time and popularity in the field of malware detection. It was intended to check its feasibility in malware classification compared to other neural network models, especially CNN. It was trained on the encoded feature sequence obtained from the autoencoder model. The MLP model acted as the final classifier in the ensemble model which trained on the output generated by the GRU model.

The proposed ensemble model, SE-AGM, was compared with other existing ensemble approaches based on the accuracy measures. The experimental outcome proved the effectiveness of our proposed architecture and showed that our approach was second to the best (IMCEC in [39] with 99.50% accuracy) with an accuracy of 99.43% on the MalImg dataset. Our proposed method was unique compared to other approaches as it combined different neural network models in an ensemble approach, which has not been carried out before. The existing works involved the excessive usage of widely available CNN-based pre-trained models. These pre-trained models were heavy-weight models which required more resources and expert knowledge for fine-tuning. Though our proposed ensemble model contained three neural network models, each of them had a simple and basic architecture, with most layer parameters set to their default values. Achieving high accuracy values with such architecture suggests that light-weight models can also be a way forward for malware detection with less resource usage, training time, and domain knowledge for fine-tuning.

The classification report for the proposed ensemble model revealed that our model had trouble classifying four types of malwares, specifically ‘Allaple.A’ and ‘Allaple.L’ as well as ‘C2LOP.gen! g’ and ‘C2LOP.P’ (especially the latter two). The similarities between different types of malwares may be the cause. According to this, to marginally boost the efficiency of our ensemble model (SE-AGM), maximizing the value of the metrics is needed. Regarding recognizing the malware classes ‘Swizzor.gen!E’ and ‘Swizzor.gen!I’, our model showed 100% precision, f1-score, and recall, whereas other neural network models had difficulty doing so.

Our suggested system performed an exceptional job of achieving all our target objectives—using a CNN-based transfer learned model trained from scratch for feature extraction, building simpler models with reduced complexity for faster training and lower resource utilization without sacrificing performance, requiring no expertise in neural networks for fine-tuning, better classification results with grayscale malware images, and a positive impact of data augmentation on malware. It outperformed many existing approaches, thereby validating our approach.

## Figures and Tables

**Figure 1 sensors-23-03253-f001:**
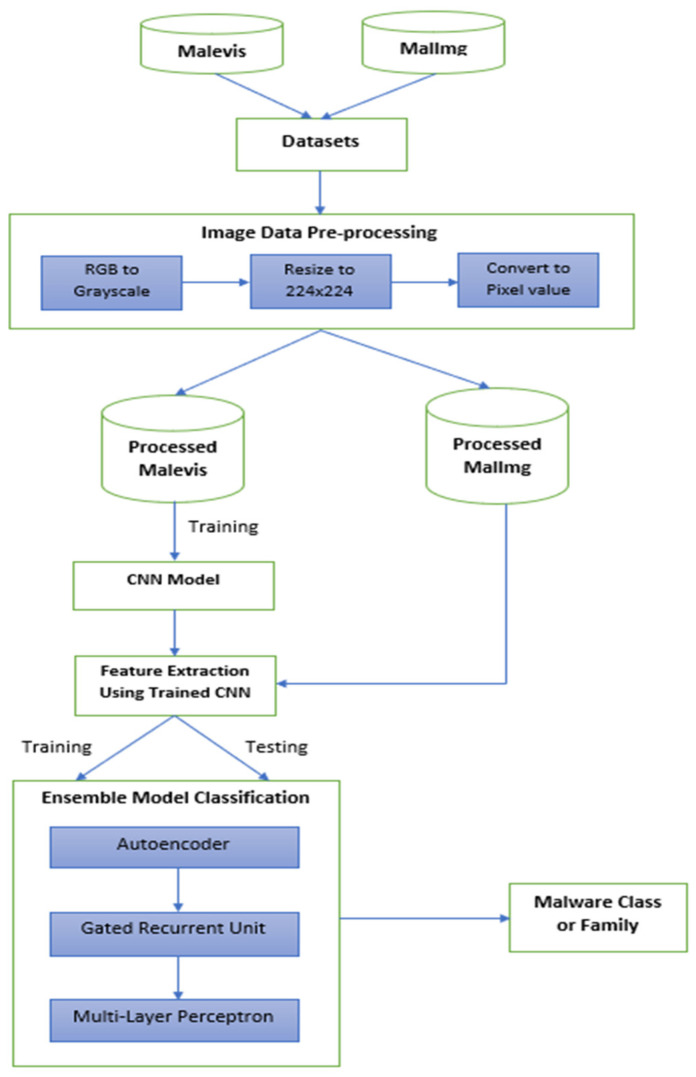
Basic architecture/work flow of the proposed system.

**Figure 2 sensors-23-03253-f002:**
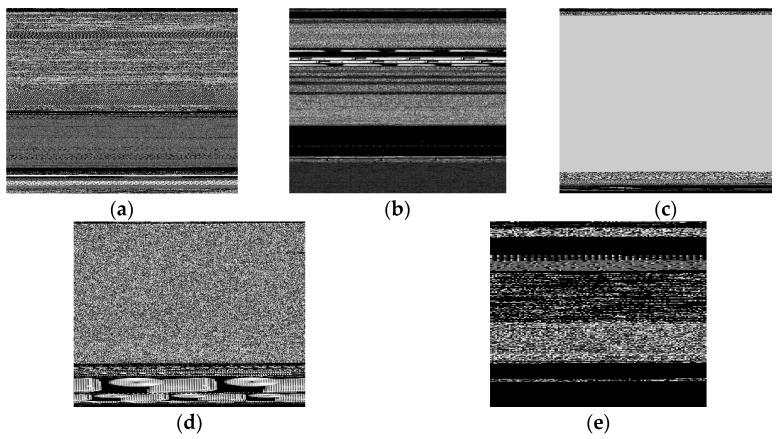
Samples of malware classes from MalImg dataset: (**a**) Adialer.C; (**b**) VB.AT; (**c**) Lolyda.AA3; (**d**) Fakerean; (**e**) Dontovo.A.

**Figure 3 sensors-23-03253-f003:**
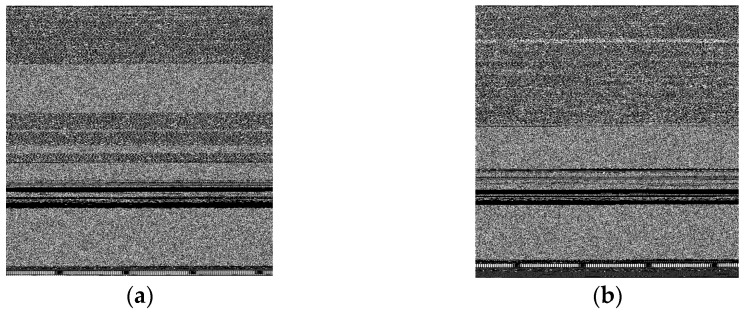
Instances of Swizzor.gen!E (**a**) and Swizzor.gen!I (**b**) class in the MalImg dataset.

**Figure 4 sensors-23-03253-f004:**
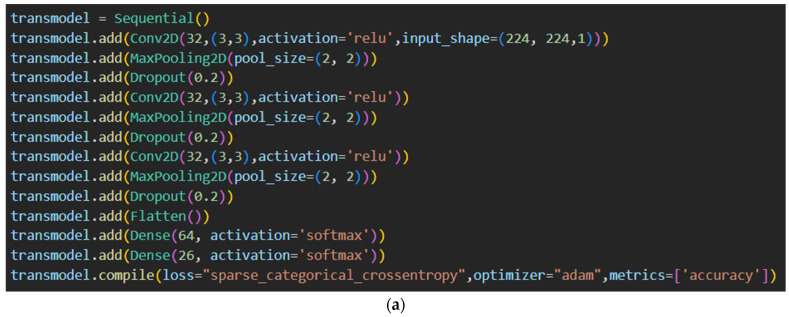
(**a**) Python code for CNN model for transfer learning; (**b**) Architecture of CNN model used for transfer learning.

**Figure 5 sensors-23-03253-f005:**
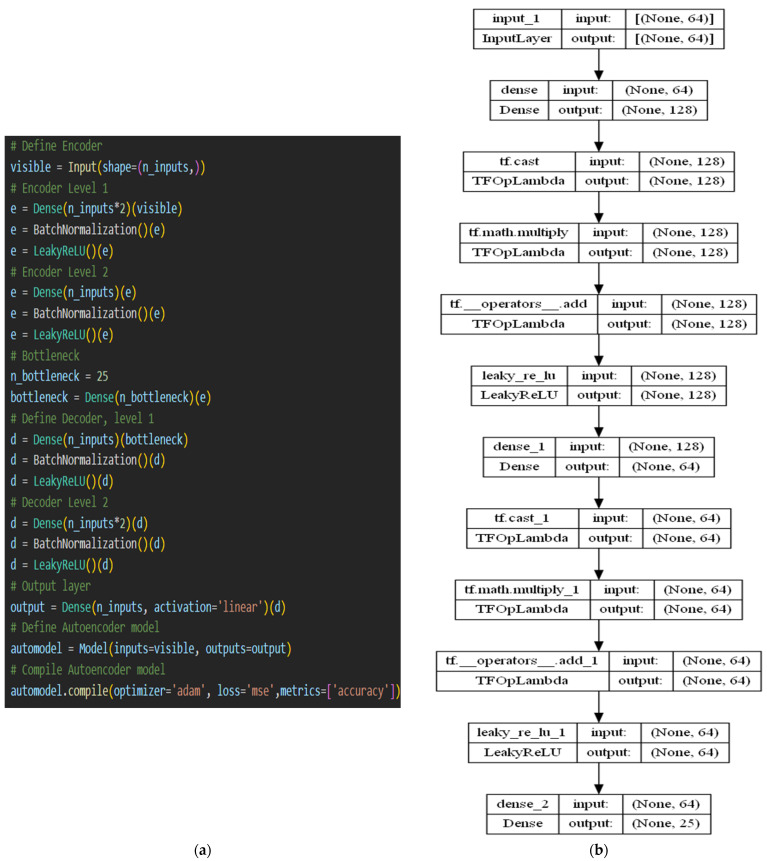
(**a**) Complete autoencoder model; (**b**) Extracted Encoder model.

**Figure 6 sensors-23-03253-f006:**
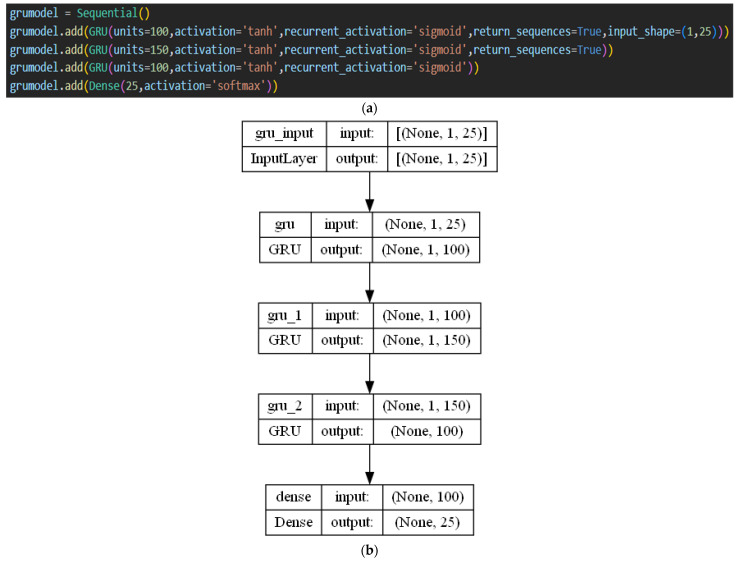
(**a**) Python code for constructing GRU model; (**b**) proposed GRU architecture.

**Figure 7 sensors-23-03253-f007:**
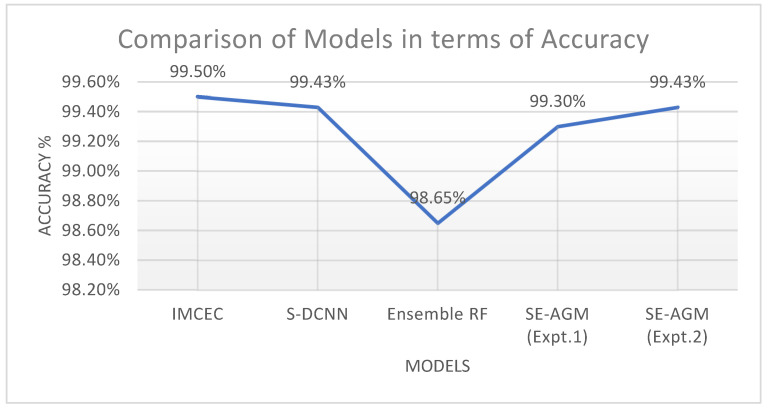
Accuracy % comparison of SE-AGM with existing models.

**Table 1 sensors-23-03253-t001:** Literature Review Summary.

Author	Publication Year	Technique	Dataset Used	Objectives	Limitations
Marastoni et al. [34]	2021	CNN, LSTM	OBF dataset (Custom dataset), MalImg, MsM2015	Effect of bicubic interpolation and custom-trained TL on malware detection.	TL average, Fixed size, obfuscation techniques, data balancing.
Casolare et al. [35]	2022	J48, LMT, RF, RT, REP Tree	Custom dataset on Android APK from Android malware repo.	Malware detection in Android applications.	No benchmark data, data balancing, not detect malware family, time diff. leads to decline.
Kim et al. [36]	2017	CNN, MLP	Microsoft Malware	Identify groups in which malware resides and detect emerging malware.	Unstable MLP, model enhancement, and data balancing.
Khan et al. [25]	2018	GoogleNet, ResNet18, 34, 50, 101, 152	Microsoft Malware, Benign software opcodes converted to images.	Use .EXE files as images for malware identification.	Heavy-weight, extensive knowledge, more execution time, large validation loss, data balancing.
Dai et al. [27]	2018	MLP, KNN, RF	VirusTotal	Malware detection using storage dump file’s content as images.	Unable to find malware in full system, hardware feature overlooked, shallow ML models, data balancing, below par accuracy.
Singh et al. [37]	2019	CNN, ResNet-50	Custom Dataset—Malshare, VirusShare, VirusTotal. MalImg	Eliminate difficulties during static and dynamic analysis by using image representation of executables.	Low for packed or unseen, obfuscation evades visualization, heavy-weight, data balancing, and undetected evasive malware.
Venkatraman et al. [38]	2019	CNN, CNN BiLSTM, CNN BiGRU	BIG 2015, MalImg	Identify malware using image-based methods and hybrid models.	Complex processing, extensive knowledge of kernels, and fine-tuning.
Vasan et al. [39]	2020	Ensemble of ResNet-50 and VGG16, SVM	MalImg, Packed malware data from VirusShare	Identification of packed and unpacked malwares.	Complex ensemble, requires extensive NN knowledge, heavy-weight, and data balancing.
Sharma et al. [40]	2020	CNN, CNN-SVM	MalImg	Maximize potential of CNN and other ML models for malware classification.	Architectural improvement, multiple SVMs for multiclass, increased model size, and data balancing.
Naeem et al. [41]	2020	Fine-tuned CNN trained on ImageNet, VGG-16, ResNet-50, Inception	MalImg, IoT-Android Mobile dataset	Develop a novel CNN-based classifier for multiclass malware classification.	Requires expert domain knowledge for fine-tuning through backpropagation.
Bakour, K., and Unver, H. M. [42]	2021	RF, KNN, DT, Bagging, AdaBoost, Gradient Boost, Ensemble Voting Classifier, ResNet, Inception-V3	Manifest.xml, DEX, Manifest-ARSC-DEX, Manifest-Resources.arsc, Manifest-ARSC-DEX-Native_jar-based image dataset.	A generic image-based classification for any file type that uses grayscale images from Android malware samples.	Static analysis, impacted by tampering and code obfuscation, injection attacks, hybrid classifier higher time.
Kumar, S. [43]	2021	TL fine-tuned CNN trained on ImageNet, ResNet-50	MalImg, BIG 2015	Refined CNN to identify unidentified malware without extensive processing and evading strategies.	Requires expert knowledge, data balancing, Uniform image size, and Common CNN-based models.
Anandhi et al. [44]	2021	DenseNet201, VGG3	MalImg, BIG 2015, Some benign samples.	To preserve the semantic information by converting malware into Markov images using Gabor filter.	Uniform image size, heavy-weight, data balancing.
Pant et al. [45]	2021	Custom CNN, VGG16, Resnet-18, Inception-V3	MalImg	To detect malware in grayscale image form.	Insufficient data, non-uniform data, pre-trained model inferior.
Kumar et al. [46]	2022	CNN trained on ImageNet, VGG16, VGG19, ResNet50, Inception V3	MalImg, Microsoft BIG	Detect malware from files obtained by converting Windows PE files into grayscale images.	Heavy-weight, data balancing, difficult fine-tuning, CNN trained on general data.
Kalash et al. [47]	2018	GIST-SVM, CNN	MalImg, Microsoft Malware	Develop a deep CNN model for malware identification using a self-learning approach.	Approach not comparable to existing solution-based on the file types, GIST-SVM not effective needs improvement.
Unver, H. M., and Bakour, K. [48]	2020	Random Forest, KNN, DT, Bagging, AdaBoost, Gradient Boost	Manifest file-based image dataset, DEX code-based image dataset, Manifest-DEX-ARSC image dataset and Android Malware Dataset.	A generic method for any type of app when converted into images for malware detection.	Static analysis methods, impacted by tampering and code obfuscation, not detect injection attacks, no data balancing.
Jin et al. [49]	2020	CNN based Autoencoders	Dataset obtained from Andro-Dumpsys study conducted by Korea University.	Detect malware using CNN-based autoencoders using uniform image size.	Uses small dataset, identifies uncollected malware as benign, separate encoder for malware, high complexity and redundancy, more resources and time.
Bakour, K., and Unver, H. M. [50]	2021	DeepVisDroid (1D CNN) trained on local and global features, 2D CNN, CNN inspired by VGG16, ResNet ad Inception-V3.	Manifest.xml file-based dataset, DEX code files-based dataset, Manifest and Resources.arsc files-based dataset, and Manifest, Resources.arsc and Dex files-based image dataset.	To detect malware by fusing deep learning techniques with image-based attributes.	High computation time, fail to acknowledge the obfuscation and camouflage used in code and commonly used pre-trained models.
Lo et al. [51]	2019	Xception model pre-trained on ImageNet, Ensemble of Xception.	MalImg, Microsoft Malware	Convert ‘bytes’ and ‘asm’ into images for malware detection using DL models.	Heavy-weight models, requires extensive knowledge, no data balancing.
Parihar et al. [52]	2022	S-DCNN (Comprising ResNet50, Xception, EfficientNet-B4) ad MLP	MalImg VirusShare	Tackle the malware detection problem using ensemble and transfer learning.	Heavy-weight ensemble model, no data augmentation.
Darem et al. [54]	2021	Ensemble—CNN and XGBoost	Small custom dataset with 9 malware types.	Detect malware using grayscale images of obfuscated opcodes present as ASM files.	No benchmark dataset, requires extensive knowledge, time-consuming.
Roseline et al. [55]	2020	Ensemble Deep Forest	MalImg BIG2015 Malevis Malicia (for validation)	Use ensemble deep forest algorithm along with a vision-based approach for high dimensional malware data.	No data augmentation.
Ding et al. [56]	2020	CNN	Dataset provided by DRE-BIN project.	Use bytecode images of malware APKs for Android malware detection.	No benchmark dataset, small data, no data augmentation, and average results.
Ngo et al. [57]	2020	Grayscale image—CNN Other features—KNN, DT, SVM, RF, etc.	IoT malware dataset by IoTPOT, IoT SOHO and VirusShare	Experimentation on existing methods of static analysis for IoT malware detection.	Average results with obfuscation and encryption, no benchmark data.
Huang et al. [58]	2021	VGG16	Malware + benign samples from ‘virussign.com’	To detect malware present in Windows OS using hybrid visualization technique.	No benchmark data, unable to identify unknown samples, average results.
Naeem et al. [59]	2020	CNN	Leopard Mobile MalImg	Using image visualization and DL models for malware detection in industrial IoT.	No data balancing, more classification time.
He et al. [60]	2019	CNN with SPP layers ResNet	Data from Andro-Dumpsys study	Assess efficacy of CNN-based model in combating superfluous API injections in malware detection domain.	SPP led to memory limitations, dataset constraints, and models not optimized.
Su et al. [61]	2018	2-layered CNN	Data from Ubuntu System files and IoTPOT dataset.	Using CNN-based approach to mitigate risks of DDoS attacks in IoT environment.	Susceptible to obfuscation, time-consuming data pre-processing.
Asam et al. [62]	2022	CNN AlexNet VGG16 ResNet50 Xception GoogleNet	IoT Malware Dataset	Develop a CNN-based model to detect malware in IoT.	Complex CNN design, time-consuming process.
Makandar, A., and Patrot, A. [63]	2017	SVM KNN	Malheur MalImg	Malware classification by using an efficient texture feature vector.	No data balancing, complex feature vector construction.

**Table 2 sensors-23-03253-t002:** An average accuracy of SE-AGM model on MalImg dataset.

Experiments (Train: Test)	Average Training Accuracy	Average Testing Accuracy
Experiment 1—70:30	99.37% approx	99.30% approx
Experiment 2—80:20	99.45% approx	99.43% approx

**Table 3 sensors-23-03253-t003:** Average metrics value computed for Experiment 1 (70:30).

Malware Classes	Average Precision		Average Recall		Average F1-Score	
	Train	Test	Train	Test	Train	Test
Adialer.C	1.00	1.00	1.00	1.00	1.00	1.00
Agent.FYI	1.00	1.00	1.00	1.00	1.00	1.00
Allaple.A	0.978	0.968	0.98	0.972	0.978	0.97
Allaple.L	0.98	0.974	0.976	0.968	0.978	0.97
Alueron.gen!J	1.00	1.00	1.00	1.00	1.00	1.00
Autorun.K	1.00	1.00	1.00	1.00	1.00	1.00
C2LOP.gen!g	1.00	0.998	0.896	0.896	0.944	0.942
C2LOP.P	0.906	0.906	0.99	0.984	0.95	0.944
Dialplatform.B	1.00	1.00	1.00	1.00	1.00	1.00
Dontovo.A	1.00	1.00	1.00	1.00	1.00	1.00
Fakerean	1.00	1.00	1.00	1.00	1.00	1.00
Instantaccess	1.00	1.00	1.00	1.00	1.00	1.00
Lolyda.AA1	1.00	1.00	1.00	1.00	1.00	1.00
Lolyda.AA2	1.00	1.00	1.00	1.00	1.00	1.00
Lolyda.AA3	1.00	1.00	1.00	1.00	1.00	1.00
Lolyda.AT	1.00	1.00	1.00	1.00	1.00	1.00
Malex.gen!j	1.00	1.00	1.00	1.00	1.00	1.00
Obfuscator.AD	1.00	1.00	1.00	1.00	1.00	1.00
Rbot!gen	1.00	1.00	1.00	1.00	1.00	1.00
Skintrim.N	1.00	1.00	1.00	1.00	1.00	1.00
Swizzor.gen!E	1.00	1.00	1.00	1.00	1.00	1.00
Swizzor.gen!I	1.00	1.00	1.00	1.00	1.00	1.00
VB.AT	1.00	0.99	1.00	1.00	1.00	1.00
Wintrim.BX	1.00	1.00	1.00	1.00	1.00	1.00
Yuner.A	1.00	1.00	1.00	1.00	1.00	1.00

**Table 4 sensors-23-03253-t004:** Average metrics value computed for Experiment 2 (80:20).

Malware Classes	Average Precision		Average Recall		Average F1-Score	
	Train	Test	Train	Test	Train	Test
Adialer.C	1.00	1.00	1.00	1.00	1.00	1.00
Agent.FYI	1.00	1.00	1.00	1.00	1.00	1.00
Allaple.A	0.99	0.994	0.98	0.976	0.986	0.986
Allaple.L	0.98	0.978	0.99	0.992	0.99	0.986
Alueron.gen!J	1.00	1.00	1.00	1.00	1.00	1.00
Autorun.K	1.00	1.00	1.00	1.00	1.00	1.00
C2LOP.gen!g	1.00	0.998	0.896	0.9	0.942	0.946
C2LOP.P	0.906	0.904	0.99	0.99	0.95	0.948
Dialplatform.B	1.00	1.00	1.00	1.00	1.00	1.00
Dontovo.A	1.00	1.00	1.00	1.00	1.00	1.00
Fakerean	1.00	1.00	1.00	1.00	1.00	1.00
Instantaccess	1.00	1.00	1.00	1.00	1.00	1.00
Lolyda.AA1	1.00	1.00	1.00	1.00	1.00	1.00
Lolyda.AA2	1.00	1.00	1.00	1.00	1.00	1.00
Lolyda.AA3	1.00	1.00	1.00	1.00	1.00	1.00
Lolyda.AT	1.00	0.998	1.00	1.00	1.00	1.00
Malex.gen!j	1.00	1.00	1.00	1.00	1.00	1.00
Obfuscator.AD	1.00	1.00	1.00	1.00	1.00	1.00
Rbot!gen	1.00	1.00	1.00	1.00	1.00	1.00
Skintrim.N	1.00	1.00	1.00	1.00	1.00	1.00
Swizzor.gen!E	1.00	1.00	1.00	1.00	1.00	1.00
Swizzor.gen!I	1.00	1.00	1.00	1.00	1.00	1.00
VB.AT	1.00	1.00	1.00	1.00	1.00	1.00
Wintrim.BX	1.00	1.00	1.00	1.00	1.00	1.00
Yuner.A	1.00	1.00	1.00	1.00	1.00	1.00

**Table 5 sensors-23-03253-t005:** Comparing the SE-AGM model with existing ensemble models on the MalImg dataset.

Models	Accuracy
IMCEC [39]	99.50%
S-DCNN [51]	99.43%
Random Forest (Ensemble) [55]	98.65%
SE-AGM (Expt. 1)	99.30%
SE-AGM (Expt. 2)	99.43%

**Table 6 sensors-23-03253-t006:** Comparison of SE-AGM with non-ensemble approaches.

Models	Accuracy
CNN [32]	98.30%
LSTM [32]	98.50%
CNN [35]	96.08%
ResNet-50 [35]	98.10%
IMCFN [39]	98.27%
MCFT-CNN [41]	99.18%
SE-AGM (Expt. 1)	99.30%
SE-AGM (Expt. 2)	99.43%

## Data Availability

All the output data have been available in the manuscript in the form of figures and tables.

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
