# Peer review of "Transfer Learning for Image-Based Malware Detection for IoT"

_sensors, 2023, doi:10.3390/s23063253_

Round 1
Reviewer 1 Report
This paper presents a study on transfer learning for image-based malware detection for IoT. The manuscript is well organized.
However, a major revision is required considering the following:
1. Describe clearly in the abstract what the original contributions in this manuscript are. Emphasize the novelty.
2. How do you justify that the proposed method will work for an arbitrary data set instead of those considered in this paper?
3. What are the limitations of SE-AGM and in general that of the current study?
4. The following relevant references should be updated: [Fuzzy Mathematics,: An Introduction for Engineers and Scientists, Springer Verlag, 2010; Soft Computing Techniques in Engineering, Health, Mathematical and Social Sciences, CRC Press, Boca Raton, 2021].
Author Response
We thank the Editor-in-Chief for giving us the opportunity to revise our manuscript. We greatly appreciate the care and diligence shown in
handing our submission. Based on the reviewers comments, we have undertaken a major revision of our manuscript.
We also thank the reviewers for their excellent feedback. These comments have helped to significantly improve the quality of the paper and
make it more readable. We greatly appreciate the thorough and thoughtful evaluation of our paper by the reviewers.

Reviewer 2 Report
Dear Authors,
Article is well structured and the topic is interesting. However, following comments should be addressed prior to further processing of the article.
1) Refer to abstract: What is SE-AGM? Each short form must be described at its first occurrence.
2) Refer to abstract: Undue use of passive voice is observed in the abstract.
3) Refer to whole article: Whole article need to be carefully revised for language checks.
4) Refer to literature review: Literature review should be complemented with a summary table comprising of study, publication year, technique, objective(s), limitation(s) etc.
5) Refer to figure 4: Figure quality is weak. It should be replaced with better quality image and rest of the figures should also be rechecked for the same.
6) Refer to line 372: Reference is missing for “The model was tested using MalImg and data obtained from VirusShare”.
7) Refer to line 372: Did authors tested / verified the proposed ensemble model accuracy with any other dataset?
8) Refer to line 397: Recheck the sentence “approaches in terms of accuracy, precision, and f-score”.
9) Refer to results: Is f1score in % or not? Recheck and update it, accordingly.
10) Refer to line 712: Mention reference with “Previous works include extensive usage of CNN, CNN-based pre-trained models”.
11) Refer to line 746: Recheck the sentence “The input and output shape of the model was set as (64,) and 64”.
12) Refer to line 749: Authors have taken MSE as loss metric. What about other loss metrics like, RMSE, MAE and MAPE etc?
13) Refer to line 850: Authors mentioned here“…mentioned earlier in section 3.5.3”. Where is section 3.5.3?
14) Refer to table 5: proposed ensemble model is compared with non-ensemble models. Why it is not compared with other ensemble model(s) of the domain? Does there exist no ensemble model for malware classification?
Good luck.
Author Response

(The authors gave the same response as above.)

Round 2
Reviewer 1 Report
now accept the paper